# Non-Invasive Iontophoretic Delivery of Cytochrome c to the Posterior Segment and Determination of Its Ocular Biodistribution

**DOI:** 10.3390/pharmaceutics14091832

**Published:** 2022-08-31

**Authors:** Laura Gisela González Iglesias, Siwar Messaoudi, Yogeshvar N. Kalia

**Affiliations:** 1School of Pharmaceutical Sciences, University of Geneva, 1211 Geneva, Switzerland; 2Institute of Pharmaceutical Sciences of Western Switzerland, University of Geneva, 1211 Geneva, Switzerland

**Keywords:** transscleral iontophoresis, cytochrome c, non-invasive ocular delivery, macromolecule, posterior segment drug delivery

## Abstract

The intact porcine eye globe model was used to demonstrate that transscleral iontophoresis could deliver a small protein, cytochrome c (Cyt c), to the posterior segment and to investigate post-iontophoretic biodistribution in the different ocular compartments. The effects of Cyt c concentration (1, 5, and 10 mg/mL), current density (3.5 and 5.5 mA/cm^2^), and duration of the current application (10 min and 1, 2, and 4 h) were evaluated. The data confirmed that transscleral iontophoresis enhanced the intraocular delivery of Cyt c under all conditions as compared to passive controls (same setup but without the current application). Increasing the Cyt c concentration resulted in a proportional enhancement in the Cyt c delivery. Increasing the current density from 3.5 to 5.5 mA/cm^2^ increased iontophoretic delivery at a Cyt c concentration of 10 mg/mL but did not appear to do so at 5 mg/mL; this was attributed in part to the effect of melanin binding. Short duration iontophoresis (10 min, 3.5 mA/cm^2^) of a 10 mg/mL Cyt c solution created a depot in the sclera. When this was followed by a 4 h incubation period, post-iontophoretic Cyt c diffusion from the sclera resulted in a different biodistribution, and Cyt c could be quantified in the posterior segment.

## 1. Introduction

Biopharmaceuticals have been established as essential therapeutics for the treatment of posterior segment diseases, including age-related macular degeneration, diabetic macular edema, and diabetic retinopathy [1,2,3]. However, their topical delivery to the back of the eye, and indeed that of low-molecular-weight drugs, poses significant challenges [4,5,6]. In addition to the static and dynamic barriers to transport imposed by the ocular tissues, the delivery of biologics presents additional difficulties due to their particular physicochemical properties, susceptibility to degradation, and specific formulation issues [1,5]. Systemic drug administration is not a realistic alternative and these formulations are usually administered by intravitreal injection [5], despite the associated risks and complications—which range from discomfort to severe complications such as bleeding, traumatic cataracts, endophthalmitis, retinal detachment, and vitreous hemorrhaging [7,8]—and the need for proximity to medical centers with qualified personnel. Topical application, followed by uptake and passive diffusion, results in poor ocular bioavailability and an inability to reach therapeutic levels in the affected posterior tissues such as the choroid and retina. It is therefore of interest to develop novel delivery systems and technologies which can enable macromolecules to reach these ocular tissues [1,5]. 

Iontophoresis is a non-invasive active drug delivery technique that can improve the transport of charged and neutral drugs into and across biological membranes. The benefits of this technique have been reported extensively for transdermal iontophoretic drug delivery [9]. However, reports have also demonstrated its potential for ocular drug delivery; transcorneal iontophoresis may allow targeted topical treatment of intracorneal diseases [10,11], while transscleral iontophoresis can enable non-invasive intraocular drug delivery [8]. The use of iontophoresis to promote ocular delivery of small molecules has been widely reported and includes riboflavin [12], carboplatin [13], ciprofloxacin hydrochloride [14], dexamethasone phosphate [15,16], methylprednisolone [17,18], methotrexate [19], and vancomycin [20]. Charged therapeutic prodrugs have been synthesized (aciclovir prodrugs [10] and triamcinolone acetonide prodrugs [21]) to benefit from electromigration. Nevertheless, the ocular delivery of proteins and peptides using this technique is limited to a few studies: bevacizumab (BVZ, 149 kDa) [22], immunoglobulin G (IgG, 150 kDa), gadolinium-label albumin (Galbumin™, 80 kDa) [23], lysozyme (14.3 kDa) [24], and cytochrome c (Cyt c, 12.4 kDa) [25].

Cyt c is a globular and water-soluble protein consisting of a single polypeptide chain. It has an isoelectric point of 10.2, and the protein contains 21 amino acids with positively charged side chains (Arg and Lys), 12 amino acids with negatively charged side chains (Asp and Glu), and three His residues [26]. It was the first protein to be delivered by transdermal iontophoresis across intact skin [27] and a subsequent report demonstrated that it could also be iontophoresed through excised ocular tissues (isolated sclera) [25]. 

The objective of the present study was to use a more complex ex vivo model—the intact porcine eye globe—to investigate postiontophoretic ocular biodistribution in the different ocular compartments, including those of the posterior segment. We have previously used the porcine eye globe to investigate the ocular biodistribution of low-molecular-weight species—amino acid ester prodrugs of acyclovir [10] and triamcinolone acetonide [21]. This proof-of-principle study aimed to demonstrate the feasibility of using transscleral iontophoresis to deliver Cyt c non-invasively to the posterior segment and to quantify the amounts delivered to the different ocular compartments.

## 2. Materials and Methods

### 2.1. Materials

Cytochrome c from equine heart (ref. C7752: MW~12.4 kDa, purity ≥ 95%), acetic acid (glacial), and Dulbecco’s Phosphate Buffered Saline (pH 7.4) were purchased from Sigma-Aldrich (Buchs, Switzerland). Sodium chloride and sodium acetate were obtained from Fisher Scientific (Loughborough, UK) and Fluka (Buchs, Switzerland), respectively.

Silver wire and silver chloride (AgCl) for the preparation of electrodes were obtained from Sigma Aldrich (Steinheim, Germany). PVC tubing (3 mm ID, 5 mm OD, and 1 mm wall) and agarose (used to prepare salt bridge assemblies) were obtained from VWR International AG (Dietikon, Switzerland) and Conda (Madrid, Spain), respectively.

The organic solvents, acetonitrile (ACN) and methanol (MeOH), were HPLC grade (Fischer Scientific, Loughborough, UK). Trifluoroacetic acid (TFA; 99% extra pure) was obtained from Acros organics (Geel, Belgium). Ultra-pure water (Millipore Milli-Q Gard 1 Purification pack resistivity > 18 MΩ.cm; Zug, Switzerland) was used to prepare all solutions. All other chemicals were at least of analytical grade. 

### 2.2. UHPLC-UV Analytical Method Development

A robust UHPLC-UV method was developed and validated for the quantification of Cyt c deposition into the different ocular tissues. The chromatographic system consisted of a Waters Acquity^®^ ultra—a high-performance liquid chromatography (UPLC^®^) core system (Baden—Dättwil, Switzerland), including a binary solvent manager, a sample manager with an injection loop volume of 50 μL, a column manager, and a PDA detector. Gradient separation was performed using a Waters Acquity UPLC^®^ Protein BEH C4 column (50 × 2.1 mm ID, 1.7 μm) fitted with a VanGuard Waters XBridge^®^ BEH C4 (5 × 2.1 mm i_d_, 1.7 µm), maintained at 35 °C. The mobile phase consisted of 0.1% trifluoroacetic acid aqueous solution (*v*/*v*) (phase A) and 0.1% trifluoroacetic acid in ACN:H_2_O (95:5 *v*/*v*) (phase B) with a flow rate of 0.15 mL/min and a runtime of 12.5 min. The injection volume was 5 μL (partial loop injection mode). MassLynx software was used for peak integration and data analysis. The peak for Cyt c was observed at 7 min using detection wavelengths of 400 nm. The LOD and LOQ were 1 and 3 μg/mL, respectively. The UHPLC-UV method was validated according to ICH guidelines. Complete details of the analytical method and its validation are provided in the Appendix A [28,29].

### 2.3. In Vitro Ocular Biodistribution of Cytochrome c

#### 2.3.1. Ocular Tissue Source

The porcine eye has been used as an ex vivo animal model due to its similar morphology to the human eye and its ready accessibility [30,31]. Porcine sclera in particular has been extensively characterized and reported as being an appropriate in vitro model for studying the ocular permeation of low- and high-molecular-weight compounds [32]. Fresh porcine eyes were obtained from a local abattoir (CARRE; Rolle, Switzerland or the Abattoir de Loëx; Bernex, Switzerland) immediately after the sacrifice of the animals (40–60 kg). To ensure the integrity of the tissues, experiments were performed within 4–8 h of harvesting. Eye globes were rinsed with saline solution to remove any remaining blood, followed by the removal of adherent muscle with a scalpel. The whole eye globes were then used for transport studies.

#### 2.3.2. Transscleral Anodal Iontophoresis

##### Experimental Setup

A custom-made diffusion cell was used to investigate the ocular distribution of Cyt c after transscleral iontophoresis [11]. The eye globe was placed on the receiver compartment, which was prefilled with 20 mL PBS; the donor compartment (permeation area of 0.6 cm^2^) was fixed over the scleral tissue close to the corneoscleral junction, using slight pressure. For the experiments investigating iontophoretic transport of Cyt c, conventional silver/silver chloride (Ag/AgCl) electrodes were used. The anode (Ag) was connected to the donor compartment containing the Cyt c aqueous solutions at the given concentrations (prepared in Milli-Q water in a volumetric flask) via saline bridges (3% agarose + 0.1 M NaCl) to reduce the competition due to electromigration of sodium ions. The cathode (AgCl) was inserted directly in the receiver compartment (PBS) through the sampling arm. The current density was applied using a constant current power generator (APH 1000 M, Kepco Inc; Flushing NY, USA). Upon completion of the experiment, the system was rapidly disassembled, and the tissue was washed with running water and cleaned with cotton swabs. Each eye globe was wrapped with parafilm and individually frozen at −80 °C for 24 h. The setup for the control experiments (passive diffusion) was the same as for the iontophoretic studies but in the absence of current application. All experiments were conducted at least in quadruplicate.

##### Effect of Concentration and Iontophoretic Parameters

The effects of (i) Cyt c concentration in the donor compartment (1, 5, and 10 mg/mL Cyt c solution in water), (ii) current density (3.5 and 5.5 mA/cm^2^), and (iii) duration of current application (1, 2, and 4 h), on the transscleral iontophoretic delivery and ocular biodistribution of Cyt c were evaluated in a series of experiments. After the initial studies into the effect of duration of current application, the feasibility of using short-duration iontophoresis (10 min) was also studied, as was how this, followed by a post-iontophoretic incubation period, affected ocular biodistribution. The different conditions evaluated are presented in Table 1.

#### 2.3.3. Post-Iontophoretic Diffusion of Cyt c

For this experiment, the donor solution was removed after iontophoretic treatment for 10 min and the setup was dismantled. The eye was cleaned with running water and with cotton swabs and then placed for 4 h in a diffusion cell. The eye surface was protected with a small plastic bag to prevent the tissue from drying out. Then, the eye globes were wrapped and frozen (−80 °C) for 24 h before further processing. 

#### 2.3.4. Tissue Preparation for Biodistribution Analysis

The eyes were dissected while frozen according to a previously reported procedure [11,33] to limit the diffusion of drugs to adjacent tissues and cross-contamination during dissection while isolating the cornea, aqueous humor, ciliary body and iris, vitreous humor, neural retina, choroid and retinal pigmented epithelium (RPE), and sclera. The tissues were weighed and separately extracted with acetate buffer (pH 3) solution for 4 h under mild agitation to extract the deposited Cyt c; the extraction volume was adjusted according to the tissue (Appendix A). Extraction samples were centrifuged at 12,000 rpm for 10 min. The supernatant was filtered using 0.22 µm cellulose acetate membrane syringe filters (Macherey-Nagel; Düren, Germany). The samples were analyzed using the validated UHPLC-UV described above. 

#### 2.3.5. Determination of the Water Content in Ocular Tissues

In order to determine the water content of the ocular tissues, the eye globes (n > 5) were dissected, and the individual tissues were placed in weighing boats. The initial weight was quickly determined, and the tissues were dried under an extraction fume hood and monitored until a constant weight was achieved. The water content (%) was then calculated.
$$\mathrm{water}\;\mathrm{content}\;\left(\%\right)=\frac{m_{i}-m_{f}}{m_{i}}\times 100$$
where m_i_ and m_f_ correspond to the initial and final weights, respectively.

#### 2.3.6. Statistical Analysis

Results of the transport experiments are expressed as mean ± standard deviation (SD). Statistical analysis was performed using Prism version 9.3.0 (GraphPad Software, La Jolla, CA, USA). Grubbs’ test, with an α of 0.05, was used to identify outliers. The results were evaluated statistically using nonparametric analysis, and a Mann–Whitney test was used to compare two data sets. The level of significance was fixed at *p* < 0.05.

## 3. Results

### 3.1. Tissue Water Content

Due to the complexity of the eye and the heterogeneous structures/compositions of each of the component tissues, different approaches have been proposed to quantify the drug deposition in biodistribution studies, i.e., consider the weight of each tissue and assume a density equal to that of water (1 g/cm^3^) [21,33], or assume equilibrium concentration with the solution [34]. In this study we determined the percent of water in the different tissues (Table 2) and used it to calculate the final extraction volume and the amounts of Cyt c deposited in each compartment.

### 3.2. Transscleral Delivery of Cytochrome c

Iontophoretic transport is influenced by the interplay between three key parameters—the concentration of the permeant, the current and the area over which it is applied (expressed as the current density), and the duration of current application [8]. These parameters were systematically varied in order to investigate their effect, not just on the absolute amounts of Cyt c that were delivered, but on the distribution in the different ocular compartments with the objective to see how duration and current density could be decreased to make ocular iontophoresis more “treatment friendly”. The approach was to establish feasibility using more “limiting” conditions and then revert to more patient-friendly/acceptable parameters. 

#### 3.2.1. Effect of Concentration

The first series of experiments was designed to investigate the effect of Cyt c concentration (1, 5, and 10 mg/mL) on its anodal iontophoretic transport and the feasibility of reaching the posterior segment non-invasively as a function of the duration of current application. In order to demonstrate the proof of principle, the first round of experiments was performed with an extreme condition: a current application of 4 h, which was then progressively decreased, and the effect on ocular biodistribution was investigated.

##### Iontophoresis of Cyt c at Concentrations of 1, 5, and 10 mg/mL (i_d_ 3.5 mA/cm^2^, Duration 4 h)

To evaluate the feasibility of using iontophoresis to deliver Cyt c non-invasively to the eye and to demonstrate the superiority of electrically assisted delivery, the data were compared to those from control conditions (i.e., passive delivery experiments performed in the absence of current). The total amounts of Cyt c delivered through the sclera after iontophoresis for 4 h—i.e., sum of the amounts of Cyt c deposited in the cornea (CO), aqueous humor (AH), iris (IR), vitreous humor (VH), retina (RE), choroid (CH), and sclera (SC)—were 853 ± 95 µg, 3568 ± 103 µg, and 7381 ± 599 µg, for the Cyt c formulations at 1, 5, and 10 mg/mL, respectively, corresponding to a 25-fold increase as compared to control for the 1 and 10 mg/mL formulations and a 40-fold increase at 5 mg/mL (Figure 1A).

Given that 1 mL of each solution was placed in the donor compartment, this implied a remarkable delivery efficiency of ~85% at 1 mg/mL and values >70% at concentrations of 5 and 10 mg/mL. These delivery efficiencies result in depletion of Cyt c from the donor compartment, meaning that there was a plateauing of the amount delivered with increasing concentration.

Figure 1B shows the biodistribution profile of Cyt c in the various ocular tissues (expressed as µg of Cyt c per g of tissue) as a function of concentration after iontophoresis for 4 h at 3.5 mA/cm^2^. A statistically significant difference was observed between the three concentrations (1, 5, and 10 mg/mL) for the SC (*p* = 0.0238), CH (*p* < 0.0476), RE (*p* = 0.0238), and VH (*p* = 0.0238).

The concentrations of Cyt c after 4 h in the RE were 43 ± 35 µg/g, 316 ± 94 µg/g, and 560 ± 136 µg/g; the corresponding values in the CH were 286 ± 74 µg/g, 980 ± 288 µg/g, and 1530 ± 321 µg/g; and for the VH were 17 ± 13 µg/g, 305 ± 49 µg/g, and 931 ± 346 µg/g from 1, 5, and 10 mg/mL solutions, respectively. Due to the characteristic red color of the Cyt c solution, its presence in the VH after iontophoresis could be confirmed by visual inspection, and it was possible to observe the increase in the intensity of the red pigmentation with respect to the Cyt c donor concentration (Figure 2).

The biodistribution profiles shown in Figure 1B demonstrated that increasing the Cyt c concentration in the donor compartment led to the presence of greater amounts in the posterior segment—e.g., the RE and VH, which are the target tissues for the treatment of several ocular pathologies using biopharmaceuticals (e.g., noninfectious intermediate, posterior and panuveitis, wet age-related macular degeneration, diabetic macular edema, and diabetic retinopathy). Only small amounts of protein were present in the CO, IR, and AH after transscleral iontophoresis. The presence of Cyt c in these regions is possibly due to lateral diffusion from the SC during prolonged iontophoresis, as well as the proximity of the formulation application site to the CO (Figure 3).

##### Iontophoresis of Cyt c at Concentrations of 1, 5 and 10 mg/mL (i_d_ 3.5 mA/cm^2^, Duration 2 h)

Since almost the entire amount of protein present in the donor compartment managed to enter the eye and avoided the depletion effect of Cyt c, it was decided to perform a second series of experiments to investigate the effect of the concentration but using a shorter 2 h duration of current application. Figure 1C shows that, after a current application of 2 h, the Cyt c delivery was >30 times higher than the control conditions (i.e., passive delivery) for concentrations of 1 and 5 mg/mL and ~50 times higher at 10 mg/mL. Delivery efficiencies were ~86% for 1 mg/mL (860 ± 88 µg), ~92% for 5 mg/mL (4578 ± 436 µg), and >60% for 10 mg/mL (6179 ± 344 µg). In agreement with the results observed for the previous condition, the increase in the Cyt c concentration from 1 to 10 mg/mL was mirrored by an increase in the total amount of Cyt c delivered to the eye globe (R^2^ = 0.90), although plateauing was again apparent.

The biodistribution profiles of Cyt c in the different segments of the eye after transscleral iontophoresis at 3.5 mA/cm^2^ for 2 h are shown in Figure 1D. The highest concentrations of Cyt c were found in the SC. The concentrations in the VH (244 ± 98 µg/g vs. <LOD (i.e., below the LOD), *p* = 0.0022), RE (240 ± 212 µg/g vs. 9 ± 10 µg/g, *p* = 0.0043), CH (1303 ± 649 µg/g vs. 97± 64 µg/g, *p* = 0.0022), and SC (2119 ± 244 µg/g vs. 428 ± 43 µg/g, *p* = 0.0022) after iontophoresis at 5 mg/mL were significantly higher than those at 1 mg/mL. However, no statistically significant difference was found in the amounts in the deep ocular tissues upon further increasing the Cyt c concentration from 5 to 10 mg/mL, with the exception of the VH (*p* = 0.0043), wherein the Cyt c deposition increased from 244 ± 98 to 651 ± 192 µg/g.

##### Iontophoresis of Cyt c at Concentration of 1 and 5 mg/mL (i_d_ 3.5 mA/cm^2^, Duration 1 h)

In the third series of experiments, we decided to further decrease the duration of the current application to 1 h; delivery efficiencies were still high, with 50% for 1 mg/mL (503 ± 226 µg) and 75% for 5 mg/mL (3742 ± 170 µg). Figure 1E shows the difference in total Cyt c delivery between the passive and iontophoretic conditions. The amounts of Cyt c delivered to the eye after iontophoresis using Cyt c solutions at 1 and 5 mg/mL were 19 and 65 times higher, respectively, than the passive controls. For the 1 h treatment (Figure 1F), the highest concentrations of Cyt c (~90%) were found in the SC, which is congruent with the reported presence of negatively charge biopolymers (hyaluronic acid, glycosaminoglycans), which contribute to the binding of positively charged molecules such as Cyt c, and slow down their transit [38]. Furthermore, melanin—present in the choroidal layer—has also been reported to bind to positively charged molecules with high affinity [25]. These binding phenomena with components of the ocular tissues have been reported to be one reason for the surprisingly poor ocular iontophoretic delivery of lysozyme, which has a similar net charge, isoelectric point (9.32), and molecular weight to Cyt c [24]. It also coincidentally mirrors Cyt c’s lack of transdermal iontophoretic delivery, in which the reduction in electroosmotic solvent has been reported to bind to fixed negative charges in the skin [39].

#### 3.2.2. Effect of Current Density

This series of experiments investigated the effect of increasing current density from 3.5 to 5.5 mA/cm^2^ on the intraocular biodistribution of Cyt c.

##### Iontophoresis of Cyt c (10 mg/mL) at a Current Density of 5.5 mA/cm^2^ for 2 h

The delivery of Cyt c at a fixed concentration and application time was measured as a function of the current density (Figure 4A). Cyt c delivery showed a statistically significant increase from 6178 ± 344 µg to 8075 ± 522 μg (*p* = 0.0043) as the current density was increased from 3.5 to 5.5 mA/cm^2^; the increase in the current density resulted in an increase in the delivery efficiency from 62% to 81%, respectively. The results also showed that the application of anodal iontophoresis at a current density of 3.5 and 5.5 mA/cm^2^ resulted in 50- and 70-fold improvements in the total delivery, respectively, over passive application (114 ± 9 μg).

The modulation of the current density did not significantly impact the biodistribution profile of Cyt c in the ocular tissues, with the exception of the CH (*p* = 0.0303), which had depositions of 1258 ± 1039 µg/g and 4362 ± 3052 µg/g at 3.5 and 5.5 mA/cm^2^, respectively (Figure 4B). The large standard deviation could be explained by the variation in the amount of melanin present in the choroidal layer (certain tissues rich in melanin appear black, whereas others, with much less, appear semi-transparent). The modulation of the current density during iontophoresis made it possible to improve not only the total quantity delivered (~80%), but more importantly, there was a decrease in the SC deposition of Cyt c as greater amounts were transported to the deeper tissues.

##### Iontophoresis of Cyt c (5 mg/mL) at a Current Density of 5.5 mA/cm^2^ for 2 h

The impact of increasing the current density from 3.5 to 5.5 mA/cm^2^ for anodal iontophoresis was also tested at a lower Cyt c concentration of 5 mg/mL. The total quantity of Cyt c delivered at 3.5 mA/cm^2^ was higher than that at 5.5 mA/cm^2^ (4578 ± 436 µg vs. 3341 ± 427 µg, respectively) (Figure 4C). This difference is statistically significant (*p* = 0.0022). This was essentially due to a much greater Cyt c deposition in the SC after iontophoresis at 3.5 mA/cm^2^ (2119 ± 244 µg/g) than at 5.5 mA/cm^2^ (1154 ± 290 µg/g). Figure 4B shows the biodistribution of Cyt c in the different tissues of the eye. However, at a higher current density, the amount deposited in the deeper tissues showed a slight increase, but statistically, the delivery in the CH, RE, and VH was equivalent. This could be attributed to the limitations of the extraction method used, particularly in the case of tissues such as the IR, CH, and RE, which contain melanin, a molecule with a high affinity for Cyt c [25]. This binding to melanin does not allow for the extraction of all the Cyt c once it penetrates into the deep tissues, and this interaction may have more impact on the quantification when a lower concentration is used (5 mg/mL).

#### 3.2.3. Effect of the Duration of Iontophoresis

The third parameter likely to affect the post-iontophoretic intraocular biodistribution of Cyt c is the duration of current application. The data from the above experiments were extracted and analyzed to investigate this further.

##### Iontophoresis of Cyt c (10 mg/mL) at 3.5 mA/cm^2^ for 2 h and 4 h

Results from the experiment using the highest concentration of 10 mg/mL and a current density of 3.5 mA/cm^2^ were used to compare the effects of iontophoresis for two fairly long periods of current application—2 (I2) and 4 (I4) h. Figure 5A demonstrates the superiority of Cyt c transscleral iontophoresis compared to the corresponding passive administrations (P2 and P4). The prolongation of the application time from 2 to 4 h increased the Cyt c delivery efficiency from 62 to 74% (*p* = 0.0028). Figure 5B shows the biodistribution of Cyt c after iontophoresis for 2 and 4 h. The deposition of Cyt c in the SC, RE, CH, and VH showed no statistically significant difference between the two time points.

##### Iontophoresis of Cyt c (5 mg/mL) at 3.5 mA/cm^2^ for 1 h, 2 h, and 4 h

The second evaluation for this parameter was performed using the data obtained at the lower concentration of 5 mg/mL (Figure 5C). After current application for 1 h, the delivery efficiency was ~75% (3742 ± 332 μg). Increasing the duration of iontophoresis to 2 h resulted in a statistically significant (*p* = 0.0043) improvement, reaching 92% (4578 ± 408 μg). However, further extension of the duration to 4 h resulted in a decrease by 20% (3568 ± 103 μg) (*p* = 0.0238). The effect of this parameter on the biodistribution profile is presented in Figure 5D. The Cyt c concentrations in the SC were 1933 ± 247, 2119 ± 244, and 1474 ± 37 µg/g, for 1, 2, and 4 h respectively. There was a significant difference between 2 and 4 h (*p* = 0.0238), again suggesting that Cyt c was penetrating deeper into the eye but was not able to be released. A statistically significant difference (*p* = 0.0152) in the concentration of Cyt c in the vitreous humor was observed between 1 and 2 h (92 ± 93 and 244 ± 98 µg/g, respectively). A nonsignificant difference was observed for the Cyt c concentrations in the CH and RE.

##### Iontophoresis of Cyt c (1 mg/mL) at 3.5 mA/cm^2^ for 1 h, 2 h, and 4 h

The last comparison was made at a concentration of 1 mg/mL. Transscleral iontophoresis resulted in a total Cyt c deposition of 503 ± 226, 860 ± 88, and 853 ± 95 µg after current applications of 1, 2, and 4 h, respectively (Figure 5E). There was a statistically significant difference between the 1 and 2 h time points (*p* =0.0087) but not between 2 and 4 h. The application of iontophoresis for 1 h resulted in the deposition of 250 ± 118 µg/g of Cyt c in the SC, but no Cyt c was detected in deeper tissues (Figure 5F). At 2 h, a significantly greater amount was deposited into the SC (*p* = 0.0087) and CH (*p* = 0.0152)—431 ± 40 and 99 ± 65 µg/g, respectively. The diffusion of Cyt c continued to the RE (7 ± 11 µg/g). After 4 h of application, the effect of duration was clear: Cyt c concentrations were significantly higher in the CH (*p* = 0.0022), RE (0.0325), and VH (*p* = 0.0152), which reached 286 ± 74, 43 ± 35, and 17 ± 13 µg/g, respectively—with the exception of the SC, which had a deposition of 566 ± 254 µg/g (Figure 5F).

In summary, in terms of the total amount of Cyt c delivered, although increasing the duration from 1 to 2 h produced a statistically significant difference, no such effect was seen when extending the duration of iontophoresis from 2 to 4 h. In parallel, the biodistribution profile showed that more Cyt c reached the deep tissues of the eye.

As described above, the prolongation of the iontophoresis duration up to 4 h seemingly resulted in a decrease in the fraction delivered. As mentioned before, this can be due to the difficulties in extracting Cyt c from these deeper tissues because of the strong affinity of Cyt c for ocular tissue components such as melanin which is present in the CH [25] and the hyaluronic acid present in the SC [24] and VH.

#### 3.2.4. Modulation of the Iontophoretic Conditions

It is obvious that the first conditions studied were extreme (e.g., duration of current application of 4 h as well as a high current densities), reaching 5.5 mA/cm^2^, and were reported to be close to the upper limit tolerated for transscleral application in humans [8]; however, it provides a first “mapping” of the ocular biodistribution of a protein under different iontophoretic conditions. Another way to visualize the ocular biodistribution of Cyt c is presented in Figure 6, which shows the percentage of the total Cyt c delivered that is deposited in the different ocular tissues. This schematic representation of the results for Cyt c concentration in the donor compartment (increasing from right to left) and the time of iontophoresis application (increasing from top to bottom), enables visualization of the delivery through the different ocular layers, which can also be a useful tool for designing and optimizing the desired application conditions to target specific tissues. The greatest proportions of Cyt c were obtained in deeper tissues with the use of a 10 mg/mL Cyt c solution and 2 h of iontophoresis (Figure 6D), with the vitreous humor as a major site of Cyt c deposition (~30%). Moreover, iontophoresis at the shortest application time (1 h) and lower concentration (5 mg/mL) resulted in appreciable deposition in the RE and VH (Figure 6B).

#### 3.2.5. Short-Duration Iontophoresis

Although the previous results were encouraging and demonstrated proof-of-principle, the conditions could not be considered as “patient-friendly” or particularly suitable for clinical applications. Therefore, since the possibility of reaching the posterior segment of the eye non-invasively had been confirmed, the aim of the next set of experiments was to investigate the Cyt c delivery and biodistribution under more realistic iontophoretic conditions.

##### Iontophoresis of Cyt c (10 mg/mL, i_d_ 3.5 mA/cm^2^) for 10 min

After a short duration iontophoresis of 10 min Cyt c delivery was 32 times superior to the passive control (Figure 7A) and enabled a total delivery of 1299 ± 326 µg, with Cyt c localized mainly in the SC (Figure 7B).

##### Iontophoresis of Cyt c (10 mg/mL, i_d_ 3.5 mA/cm^2^) for 10 min Followed by Passive Diffusion for 4 h

The same conditions were then tested, followed by 4 h of passive diffusion to investigate the post-iontophoretic diffusion of Cyt c from the scleral depot. The total delivery of Cyt c was similar and 13% of the amount delivered was found in the SC—whereas after 4 h of post-iontophoretic diffusion, the protein was also found in the CH and was detected in the RE and VH (Figure 7C). 

## 4. Discussion

In this study, the deposition and distribution of Cyt c in the different ocular tissues was determined after the application of different iontophoretic conditions. Cyt c was chosen due to its demonstrated ability to be iontophoresed efficiently into biological tissue [25,27]. The biodistribution profiles obtained were used to establish the feasibility of reaching posterior segment tissues.

The SC is considered to be more permeable than the CO, and is formed by collagen, proteoglycans, and elastin chains that create a fiber matrix containing aqueous channels [40,41,42]; the average effective radius of these channels/pores in the SC have been estimated to be ~10–40 nm [43], which allows for the penetration of large molecules (<150 kDa) [40,44]. These properties have motivated research into the use of transscleral drug delivery as a means of reaching the back of the eye. The arrangement of collagen and elastin fibers that constitute the scleral matrix and the presence of negatively charged proteoglycans affect the rate of drug diffusion. The principal drug-related factors that determine transscleral permeation are the molecular weight and radius, lipophilicity, and net surface charge [42,45]. However, recent reports also highlight the importance of macromolecule-related factors such as water solubility, shape, conformation, and surface charge distribution.

Interactions with water-soluble components present in the scleral matrix, i.e., glycosaminoglycans, dermatan sulfate, chondroitin sulfate, and hyaluronic acid, can affect the diffusion of molecules through the SC [24]. Transport can also be influenced by the subjacent layers, i.e., CH and RE, which can also restrict drug diffusion. The passive permeation of Cyt c through the “trilayer” composed of the sclera–choroid–Bruch’s membrane and isolated SC have been compared and a five- to seven-fold reduction in the amount of protein permeated was observed in the presence of the CH and Bruch’s membrane [25]. This reduction was attributed to the binding of Cyt c to the melanin present in the CH. Reduction in permeation due to interactions with melanin have also been reported for oligonucleotides [46], triamcinolone acetonide [21], celecoxib [47], and timolol [48].

Despite the interaction of Cyt c with the ocular tissues, the use of iontophoresis increased Cyt c delivery 25- to 65-fold for the different conditions tested and 32-fold when applying more clinically relevant conditions (3.5 mA/cm^2^ for 10 min). However, superficially similar proteins can display very different iontophoretic transport behavior—e.g., lysozyme (14.3 kDa, pI = 9.32 and a net charge of +8 at pH 7.4), which even had a higher electrophoretic mobility than Cyt c, displayed much poor transdermal iontophoretic permeation [39], and a had 9-fold lower transscleral iontophoretic delivery [24]. The difference was explained by the presence of stronger electrostatic and hydrophobic interactions of lysozyme with the ocular tissue and the differences in the shape of the molecules (prolate spheroid vs. spherical shape of Cyt c) that influence their diffusion in the SC pores [24].

The transdermal iontophoresis of Cyt c at two concentrations (0.35 mM and 0.7 mM in ultrapure water) was not statistically different, in agreement with the electrodiffusion model, wherein, in the absence of competing cations, the iontophoretic flux is independent of the donor concentration and depends only on the mobility of the cation and the main counterion (Cl- from the tissue) [27]. In the present study, Cyt c was the only ion present in the donor compartment, but here, increasing its concentration in the donor compartment resulted in an increase in the total delivery of the protein for all the conditions tested (1, 2, and 4 h application time). A previous report studying the iontophoretic delivery of Cyt c through isolated porcine SC showed the same tendency (conditions: 5 and 10 mg/mL, 2.9 mA/cm^2^ for 2 h) [25].

Transscleral iontophoresis resulted in exceptional Cyt c delivery efficiencies of ~85% (1 mg/mL, 4 h), >70% (5 and 10 mg/mL, 4 h), ~86% (1 mg/mL, 2 h), and ~92% (5 mg/mL, 2 h), so it is reasonable to hypothesize that the increase in Cyt c concentration also increases the Cyt c delivery since it offsets the effect of protein depletion from the donor compartment (given the high fraction of protein load permeated across the SC). The higher Cyt c delivery achieved by using iontophoresis is explained by the fact that electromigration is the main mechanism responsible for Cyt c transport [25,27] and not electroosmosis, as is the case of other macromolecules such as bevacizumab [22] and dextrans [33].

In general, the current density can be modulated to reduce the possible side effects of transscleral iontophoresis, such as irritation of the RE. The current densities selected in this work correspond to the reported safe conditions tested in human SC [8]. An increase in this parameter from 3.5 to 5.5 mA/cm^2^ showed a clear positive effect in the total Cyt c delivered from a 10 mg/mL solution into the SC (6178 ± 344 µg vs. 8075 ± 522 µg), and a slight increase in the CH deposition. The proportionality between this parameter and the scleral iontophoretic permeation of Cyt c was also reported by Tratta et al. [25], using isolated porcine scleral tissue and increasing the current density from 1.5, 2.9, and 5.8 mA/cm^2^. The effect of this parameter on the delivery and ocular biodistribution of IgG (150 kDa) has been explored in New Zealand rabbits using a commercial iontophoretic applicator (Visulex-I). A 25 mg/mL IgG solution (low ionic strength = 0.0015 M) was applied at 1.86 and 3.6 mA/cm^2^ for 20 min—once again, the increase in the current density produced a 1.6-fold increase in the amount of IgG delivered into the eye (438.0 ± 63.3 µg vs. 727.1 ± 37.3 µg). The application of the higher tested current density deposited ~80% of the total delivered amount in the SC and conjunctiva, ~6% and <1% reaching the RE/CH and VH, respectively [23].

Another key factor influencing the iontophoretic drug delivery is the duration of the current application. From the biodistribution mapping that was presented previously, it is possible to clearly visualize the effect of the application time in the intraocular biodistribution of the protein. When the application time increased from 1 to 2 h, a greater fraction of the Cyt c fraction was delivered (at 5 mg/mL, increasing from 75% to 92% and at 1 mg/mL, increasing from 50% to 85%). Greater amounts of Cyt c were measured in the SC, CH, RE, and even in the VH. Increasing the duration of iontophoresis from 2 to 4 h resulted in a higher total protein delivery from a 10 mg/mL Cyt c donor concentration. The biodistribution profiles showed a slight increase in the penetration of the protein into deeper tissues; however, the dissection and Cyt c extraction procedure increased the variability, making it difficult to identify statistical differences to drive a conclusion.

The use of short-duration iontophoresis of 10 min—which is more appropriate for clinical applications—led to a 32-fold increase in Cyt c deposition compared to the passive application. Experiments using an excised SC have demonstrated that the permeation of Cyt c after iontophoresis returned rapidly to the passive value when the current was interrupted and that after an initial loading of protein in the scleral tissue it was possible to quickly modify the Cyt c permeation flux by alternating different iontophoretic conditions and passive permeation [25]. The biodistribution profile of Cyt c after iontophoresis for 10 min in combination with 4 h of passive diffusion showed that it was possible to create a drug reservoir in the SC that enabled a post-iontophoretic sustained release of Cyt c, reaching deeper ocular tissues, CH and RE, and small amounts of protein were even present in the VH.

Scleral biodistribution studies were carried out after the application of a 30 min anodal iontophoresis of fluorescein isothiocyanate labelled bevacizumab (FITC-BVZ), with an ex vivo human SC using a concentration of 2.5 mg/mL at pH 7.4 (when the antibody is essentially uncharged) at a current density of 3.8 mA/cm^2^. The results showed that FITC-BVZ was deposited deeper into the scleral structure, while passive delivery only enabled the molecule to reach the superficial spaces between collagen fibers [22].

The effect of clearance after cathodal iontophoretic delivery (100 mg/mL, 4 mA, 20 min) of Galbumin™ was reported in New Zealand white rabbits using a 3-T MRI system. Thirty minutes after transscleral iontophoresis, the deposition in the conjunctiva/superior muscle was 28 ± 9 µg, and in the SC/CH/RE was 13 ± 4 µg; after 12 h, the amounts decreased to 3.7 ± 0.8 µg and 2.7 ± 2.5 µg, respectively. There was no presence of the macromolecule in the anterior chamber or VH [49]. The results obtained in the present study using the whole eye globe model are promising but it is necessary to keep in mind that the model lacks the effect of dynamic barriers that are present in vivo (i.e., clearance through blood and lymphatic vessels, bulk fluid flow, and active transport mechanisms of RPE transport proteins).

## 5. Conclusions

Transscleral delivery is a multistep process beginning with the entry of the molecule into the SC from the donor and followed by its transport across several layers of tissue (bilayer choroid–Bruch’s and retinal pigment epithelium (RPE)) before it reaches the RE and finally the VH. It is also impacted by the presence of melanin, hyaluronic acid, and other ocular components capable of binding permeant molecules with high affinity. The results presented here demonstrate that iontophoresis can be used to address some of these obstacles; iontophoretic delivery of Cyt c under certain conditions was up to 54-fold greater than that obtained after passive diffusion and the fraction delivered ranged from 50% to 92% of the applied dose (depending on the conditions). Applying a higher current density for a longer period appeared to be more effective in maximizing the amounts of protein in the VH and RE. The maximum loading of Cyt c into the SC was reached after 2 h; a further increase in the application time did not result in a higher total delivery or a higher deposition on the SC but in a higher diffusion of Cyt c to reach deeper tissues (VH).

Given the lack of previous reports on ocular biodistribution, the aim of this study was first and foremost to act as a proof of concept to show how the intact eye globe model could be used to determine post-iontophoretic ocular biodistribution and thereby demonstrate that it was possible to deliver a protein non-invasively to the posterior segment. It is clear that much further work needs to be done, but the use of transscleral iontophoresis could constitute a feasible alternative to conventional invasive treatments for some diseases of the posterior segment. The parameters discussed above demonstrate how the modulation of transscleral iontophoretic parameters could be used to modulate ocular biodistribution. The positive results obtained here with Cyt c support the transposition of this technique to therapeutic proteins with the appropriate physicochemical properties. However, it is important to reiterate the lack of dynamic barriers in our model—the directed diffusion of molecules due to iontophoresis may mean that they are less susceptible to the impact of dynamic barriers than after passive administration.

## Figures and Tables

**Figure 1 pharmaceutics-14-01832-f001:**
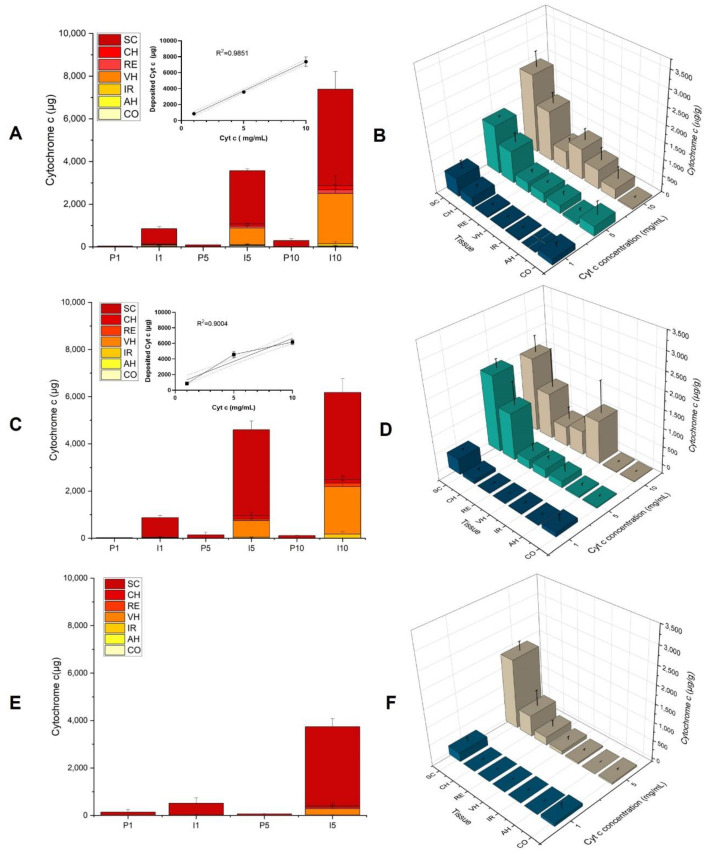
Total Cyt c delivery after iontophoresis (I) of solutions containing 1, 5, and 10 mg/mL of protein (I1, I5, and I10) at a current density of 3.5 mA/cm^2^ for (**A**) 4 h, (**C**) 2 h, and (**E**) 1 h, and comparison with results after passive delivery under the same conditions (P1, P5, and P10). The results are presented as cumulative stacked plots of the amounts in the individual tissue. Direct comparison of the Cyt c biodistribution profile in the different ocular tissues after iontophoresis at 3.5 mA/cm^2^ for (**B**) 4 h, (**D**) 2 h, and (**F**) 1 h, demonstrates how increasing the duration of current application enables deeper penetration and the vitreous humor to be reached. The values for Cyt c biodistribution are presented as the amount of Cyt c extracted per gram of tissue. Results are expressed as the mean ± SD.

**Figure 2 pharmaceutics-14-01832-f002:**
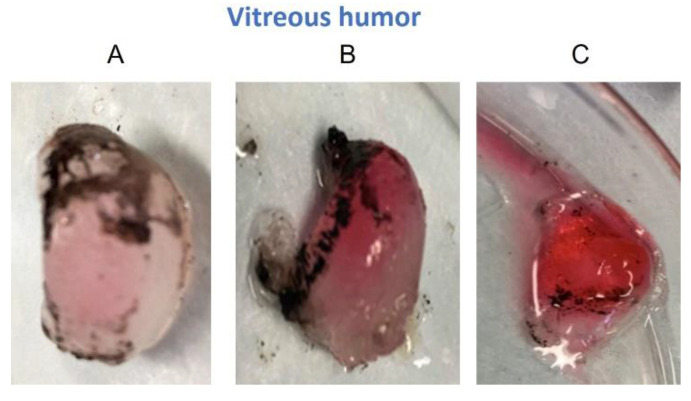
The quantitative determination of the increasing amounts of Cyt c in the vitreous humor upon increasing the Cyt c concentration in the donor compartment from 1 to 5 and 10 mg/mL could be corroborated by simple visual inspection of the red color observed in the vitreous humor after iontophoresis ((**A**) (1 mg/mL), (**B**) (5 mg/mL), and (**C**) (10 mg/mL)).

**Figure 3 pharmaceutics-14-01832-f003:**
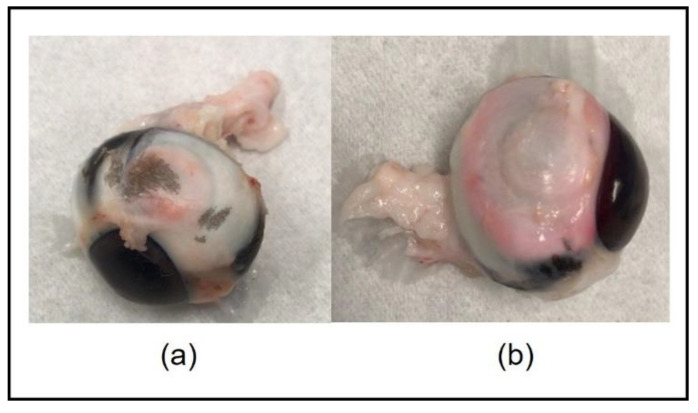
Eye globes after (**a**) passive and (**b**) iontophoretic delivery demonstrate the presence of greater lateral diffusion after current application.

**Figure 4 pharmaceutics-14-01832-f004:**
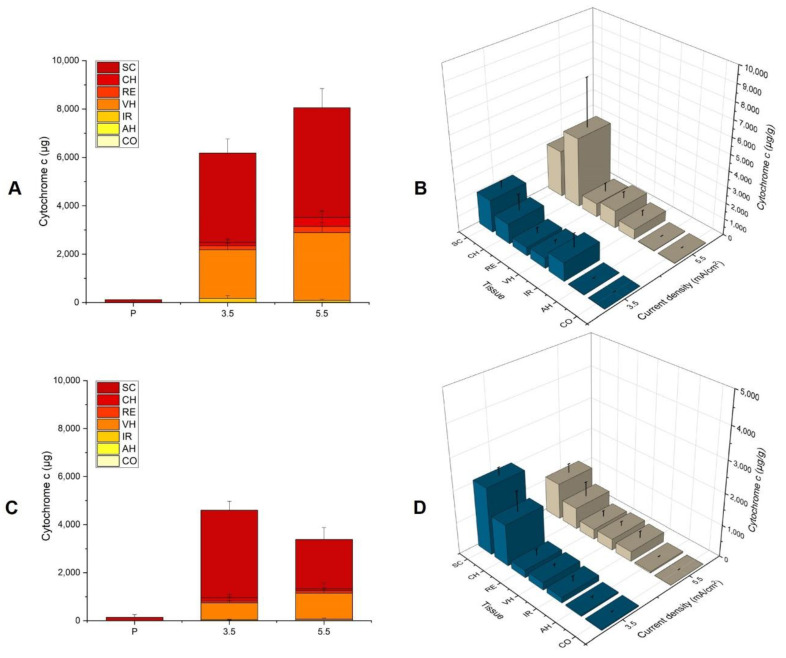
Total Cyt c delivery after iontophoresis at 3.5 or 5.5 mA/cm^2^ for 2 h using solutions containing (**A**,**B**) 10 mg/mL and (**C**,**D**) 5 mg/mL protein and comparison with passive delivery (P). Direct comparison of the Cyt c biodistribution profile in the different ocular tissues after iontophoresis (current density 3.5 or 5.5 mA/cm^2^) for 2 h using (**B**) 10 mg/mL and (**D**) 5 mg/mL protein concentrations. The values are presented as the amount of Cyt c extracted per gram of tissue. Results are expressed as the mean ± SD.

**Figure 5 pharmaceutics-14-01832-f005:**
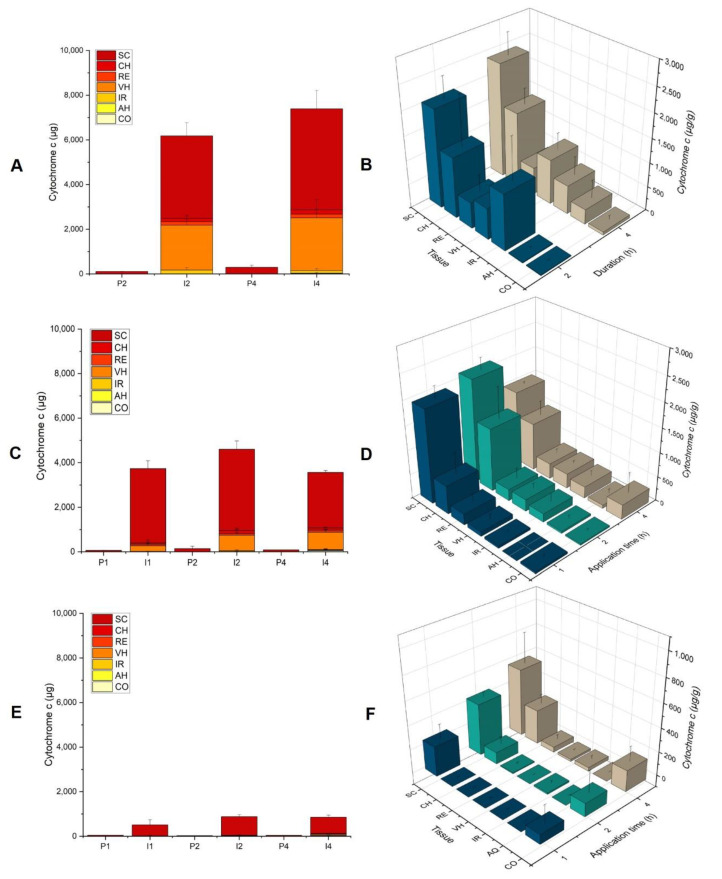
Effect of the duration of iontophoresis: Total Cyt c delivery after iontophoresis (I) at a current density of 3.5 mA/cm^2^ for 1, 2, and 4 h (I1, I2, and I4) of solutions with concentrations of (**A**) 10 mg/mL, (**C**) 5 mg/mL, and (**E**) 1 mg/mL Cyt c and comparison with results after passive delivery under the same conditions (P1, P2, and P4). Direct comparison of the Cyt c biodistribution profile in the different ocular tissues after iontophoresis at 3.5 mA/cm^2^ for 1, 2, and 4 h, using Cyt c concentrations of (**B**) 10 mg/mL, (**D**) 5 mg/mL, and (**F**) 1 mg/mL. The values for biodistribution are presented as the amount of Cyt c extracted per gram of tissue.

**Figure 6 pharmaceutics-14-01832-f006:**
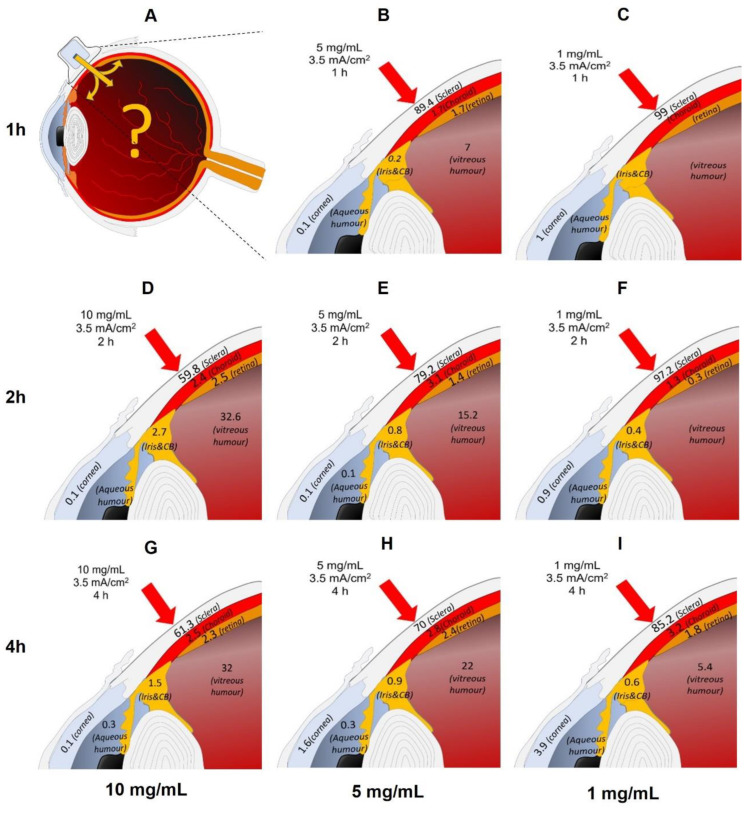
Effect of varying iontophoretic conditions (**A**). Concentration (increasing from right to left) and time of application (increasing from top to bottom) on the ocular biodistribution profile of Cyt c after transscleral iontophoresis. The numerical values given, (e.g., in (**B**), 89.4 (sclera)) represent the percentage of the total amount of delivered Cyt c found in that tissue (i.e., for this condition, 89.4% of the Cyt c delivered to the eye was recovered from the sclera).

**Figure 7 pharmaceutics-14-01832-f007:**
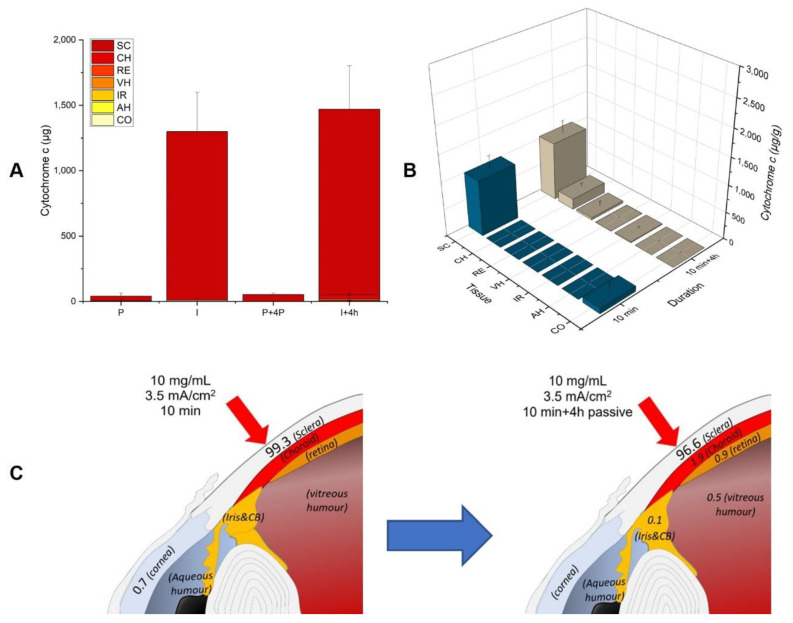
(**A**) Total delivery of Cyt c (µg) after iontophoresis (10 mg/mL, i_d_ 3.5 mA/cm^2^) for 10 min (I) and for 10 min followed by 4 h passive diffusion (I + 4P) with the respective passive controls (P, P + 4P). (**B**) Comparison of Cyt c biodistribution profile in the different eye tissues after 10 min iontophoresis and after 10 min iontophoresis followed by 4 h passive diffusion. The values are presented as the amount of Cyt c extracted per gram of tissue. (**C**) Schematic representation of Cyt c distribution. As in Figure 6, the values represent the percentage of the total amount of Cyt c delivered that was found in that tissue.

**Table 1 pharmaceutics-14-01832-t001:** Iontophoretic and formulation parameters tested for their effect on transscleral delivery and ocular biodistribution of Cyt c ^a^.

[Cyt c] (mg/mL)	Duration ^b^ (h)	Current Density (mA/cm^2^)
1	1, 2, 4	3.5
5		
10	2, 4	
5	2	5.5
10		
10 ^c^	10 min and	3.5
	10 min + 4 h passive diffusion	

^a^ Passive controls were performed using the same setup as the iontophoretic experiments but in the absence of current application. ^b^ Duration given in hours unless indicated otherwise. ^c^ Current application for 10 min was followed by either immediate freezing or incubation for 4 h to enable passive diffusion and redistribution.

**Table 2 pharmaceutics-14-01832-t002:** Water content in ocular tissues.

Tissue	Experimentally Determined Water Content (%)	Reported Water Content (%)
Cornea (CO)	71.3 ± 6.9	72–78 [30,35]
Aqueous humor (AH)	99 ± 0.3	NR
Iris and Ciliary body (IR)	58.8 ± 7.2	NR
Retina (RE)	92.4 ± 0.5	NR
Choroid (CH)	70 ± 2.5	NR
Sclera (SC)	61.4 ± 6.2	65–75 [30,36]
Vitreous humor (VH)	99.2 ± 0.3	98–99.7 [37]

NR—not reported.

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
