# Peer review of "Non-Invasive Iontophoretic Delivery of Cytochrome c to the Posterior Segment and Determination of Its Ocular Biodistribution"

_pharmaceutics, 2022, doi:10.3390/pharmaceutics14091832_

Round 1
Reviewer 1 Report
Non-invasive iontophoretic delivery of cytochrome c to the 2 posterior segment and determination of its ocular 3 biodistribution
Manuscript ID: pharmaceutics-1839735-peer-review-v1
The authors utilized a small protein, cytochrome c (Cyt c), to demonstrate transscleral delivery of biologic molecules to the posterior section of the eye. The authors used iontophoresis for the delivery of Cyt c. The authors also investigated the post-iontophoretic biodistribution of Cyt c in the different ocular compartments. The authors reported that t transscleral iontophoresis enhanced intraocular delivery of Cyt c under all conditions as compared to passive controls. The authors have demonstrated an efficient an non-invasive method of delivering large molecules like anti VEGF antibodies to the back of the eye. The conclusions are supported by an extensive and descriptive study design. All the ex vivo experiments conducted by the authors are perfect. The description and the conclusion sections perfectly describe the results and the implications of this research for ocular drug delivery.
The manuscript is perfectly drafted. The study section is quite extensive and answers all the questions about iontophoresis delivery of Cyt c. Hence this manuscript can be accepted without any changes.
Reviewer 2 Report
1. Line 22: It would be better to use the term “posterior segment” rather than “interior”.
2. The SI unit of time should be mentioned as “h” rather than “hour” throughout the manuscript. At some places for time the author mentioned “minutes” while at other places “min”, mention it uniformly.
3. Rewrite the sentence, Line 30-32: “In addition to the static and dynamic barriers to transport imposed by the ocular tissues, biologics present difficulties due to their physicochemical properties, susceptibility to degradation, and specific formulation issues [1, 5]”. It is not clear.
4. How the present study is different from the previously reported study “Tratta, E., et al., In vitro permeability of a model protein across ocular tissues and effect of iontophoresis on the transscleral delivery. European Journal of Pharm. Biopharm. 2014. 88(1): p. 116-22”?
5. What are these values indicating as mentioned in the Figure 6 and 7 (different parts of eyes with some values without any specific units, such as 70 (sclera), 1 (cornea) and so on)?
6. Rewrite the sentence mentioned in line 83-84.
7. Line 99: if the peak for Cyt c was observed at 7 min, then why the total run time was 12.5 min? If this is the actual run time, then why not HPLC should be used for the detection of the analyte in place of UPLC?
8. What was the concentration range for the calibration curve? How the LOD and LOQ values were too much higher? The method validation lacks the use of any internal standard, while Cyt c was detected in biological samples where, there were chances of interference in peak resolution due to matrix effect?
9. Line 111: Mention approximate time rather than “a few hours” after harvesting.
10. Line 138-140: “The feasibility of using short duration iontophoresis (10 min) was also studied as was how this followed by a post-iontophoretic incubation period affected ocular biodistribution”, correct the sentence. Why not the short duration iontophoresis (10 min) was also included at line 137? Why it was mentioned separately?
11. The version of GraphPad Prism is missing in section 2.3.6. Statistical analysis.
12. What was the control as mentioned in line 223?
13. Line 263: “several ocular pathologies using biopharmaceuticals”, what are such pathologies?
14. Line 286-287: “The concentrations in the vitreous humor 285 (244 ± 98 µg/g vs <LOD), what does it mean? What are the values associated (vs) with the values mentioned in Line 286-287? Check the units of these values?
15. Line 299-300: “The amount of Cyt c delivered to the eye at 1 and 5 mg/mL was 19- and 65-fold higher, respectively”, the folds higher were with respect to what?
16. As the abbreviations for the cornea, iris, aqueous humor etc were already mentioned then only abbreviations should be used through the manuscript. For examples as presented in line 264 and other places too.
17. Line 579-580: “and in the sclera, choroid and retina was 13 ± 4 µg; after 12 h the amounts decreased to 3.7 ± 0.8 and 2.7 ± 2.5, respectively”, there should be three values in Line 579 and the units were missing with values mentioned in Line 580.
18. The main intention (or indication) to deliver the Cyt c to the posterior segment of eye is missing either in the abstract or in the introduction of this manuscript.
Reviewer 3 Report
The manuscript entitled “Non-invasive iontophoretic delivery of cytochrome c to the posterior segment and determination of its ocular biodistribution” is well designed and written well. But the manuscript needs English language and grammar edits. Manuscript needs to be revised as per the suggested edits also.
In abstract, write the positive controls used for the studies.
The enhancement of Cyt c with 5 mg/mL was reduced due to melanin binding. But there is no proof of experiment for this mechanism. I strongly suggest that conduct the in vitro melanin binding effect of Cyt c with different concentrations and find out the optimal concentration of the Cyt c for melanin binding activity.
Further, the increased current effect also not proportionally correlated with the concentration of Cyt C. How the effect of higher level of current on the iontophoretic delivery should be discussed.
Write the significance and role of Cyt c in ocular diseases. How the intophoretic delivery of Cyt c beneficial compared with the other delivery methods (implants, solution, systemic administration).
In materials - Write the pH of the DPBS solution, Batch number of the Cyt C with purity.
UHPLC – use UFLC or HPLC in the manuscript. It is little confusing to the readers.
Write the preparation method of Cyt c solutions.
Results section – write the statistical testing of the data.
Conclusion of the manuscript very lengthy. Write in a brief and concise model.
Round 2
Reviewer 2 Report
The manuscript has been revised as per the suggestions. Now it looks better and suitable to be published in this journal.
Reviewer 3 Report
No further comments.